# Effects of infection history on dengue virus infection and pathogenicity

Tim K. Tsang [1], Samson L. Ghebremariam[1], Lionel Gresh [2], Aubree Gordon [3], M. Elizabeth Halloran[4,5], Leah C. Katzelnick[6], Diana Patricia Rojas[1], Guillermina Kuan[7], Angel Balmaseda[8], Jonathan Sugimoto[4], Eva Harris [6], Ira M. Longini Jr.[1,9] & Yang Yang[1,9]

The understanding of immunological interactions among the four dengue virus (DENV) serotypes and their epidemiological implications is often hampered by the lack of individual-level infection history. Using a statistical framework that infers full infection history, we analyze a prospective pediatric cohort in Nicaragua to characterize how infection history modulates the risks of DENV infection and subsequent clinical disease. After controlling for age, one prior infection is associated with 54% lower, while two or more are associated with 91% higher, risk of a new infection, compared to DENV-naive children. Children >8 years old have 55% and 120% higher risks of infection and subsequent disease, respectively, than their younger peers. Among children with ≥1 prior infection, intermediate antibody titers increase, whereas high titers lower, the risk of subsequent infection, compared with undetectable titers. Such complex dependency needs to be considered in the design of dengue vaccines and vaccination strategies.

[1] Department of Biostatistics, College of Public Health and Health Professions, University of Florida, Gainesville, FL 32611, USA. [2] Sustainable Sciences Institute, Managua 14007, Nicaragua. [3] Department of Epidemiology, School of Public Health, University of Michigan, Ann Arbor, MI 48109, USA. [4] Vaccine and Infectious Disease Division, Fred Hutchinson Cancer Research Center, Seattle, WA 98109, USA. [5] Department of Biostatistics, University of Washington, Seattle, WA 98195, USA. [6] Division of Infectious Diseases and Vaccinology, School of Public Health, University of California, Berkeley, CA 94720, USA. [7] Centro de Salud Sócrates Flores Vivas, Ministry of Health, Managua 12014, Nicaragua. [8] Laboratorio Nacional de Virología, Centro Nacional de Diagnóstico y Referencia, Ministry of Health, Managua 16064, Nicaragua. [9] Emerging Pathogens Institute, University of Florida, Gainesville, FL 32610, USA. These authors contributed equally: Tim K. Tsang, Samson L. Ghebremariam, Lionel Gresh, Aubree Gordon. These authors jointly supervised this work: Eva Harris, Ira M. Longini, Jr., Yang Yang. Correspondence and requests for materials should be addressed to E.H. (email: eharris@berkeley.edu) or to I.M.L. Jr. (email: ilongini@ufl.edu) or to Y.Y. (email: yangyang@ufl.edu)

As a leading mosquito-borne infectious agent, dengue virus (DENV) infects up to 390 million people annually worldwide, 25% of whom suffer from clinical disease. Dengue epidemics have been expanding from tropical to subtropical regions in recent decades and now put 3.9 billion people at risk, partly fueled by urbanization and travel[1–3]. With four antigenically distinct, but immunologically cross-reactive serotypes (DENV-1–DENV-4), dengue has one of the most complex transmission processes in human populations among all infectious diseases. It is widely accepted that an infection with any serotype offers long-term, if not life-long, immunity to disease due to that serotype, but only short-term heterologous immunity to other serotypes. In addition, a second infection is more likely to present with severe symptoms such as dengue hemorrhagic fever (DHF) or dengue shock syndrome (DSS) than a primary infection[4–6]. However, how exactly previous infection history modulates the risk of subsequent infection outcome is not entirely clear, partly because of the difficulty in determining retrospectively the complete infection history of individuals using current technology.

The Pediatric Dengue Cohort Study (PDCS) is an ongoing longitudinal study conducted in Managua, Nicaragua[7]. In this study, a prospective cohort of about 3800 children aged 2–9 were initially enrolled in 2004. Enrollment continued in subsequent years and the age span of the cohort was extended to 2–14 years. Study participants are encouraged to seek medical care at a study health center for all illnesses. Clinical specimens are collected from children who meet the 1997 World Health Organization (WHO) suspected dengue case definition or have undifferentiated fever, and are evaluated using reverse transcriptase polymerase chain reaction (RT-PCR), an in-house IgM capture enzyme-linked immunosorbent assay (MAC-ELISA), and an inhibition ELISA (iELISA) for acute DENV infection. To ascertain inapparent DENV infections, blood is drawn annually from each participant. For the annual serum samples, the iELISA is used to detect nonserotype-specific DENV infections. In addition, a plaque reduction neutralization test (PRNT, before 2007) and a neutralization titration (NT) assay using reporter virus particles are used to characterize serotype-specific neutralizing antibody (Nab) responses on a non-random subset of iELISA-positive samples, mostly from children with repeated infections based on the iELISA.

Combining enhanced passive case detection and annual serology tests, this large cohort study provides one of the most comprehensive data sets for investigating dengue transmission and the effects of infection history on both the risk of infection and the risk of developing dengue disease given infection in an endemic country. We refer to the risk of disease given infection as pathogenicity, as often seen in the epidemiological literature, although pathogenicity also has the implication of severe disease in other fields[8]. Specifically, the study provides clues to the following questions: (1) what are the serotype-specific annual infection risks during the study years? and (2) what are the effects of infection history and other host factors on the risk of infection and the risk of developing symptoms given infection? The estimation of infection risk differs from the estimation of disease incidence as the latter is usually based on syndromic surveillance data and does not account for inapparent infections and reporting bias. Serotype-specific forces of infection (FoI) were estimated for Iquitos, Peru from 1999 to 2010 by applying a smoothing splines approach to several longitudinal serology cohorts[9]. However, the estimation was done under the assumption that the infection processes of the four serotypes were independent, and the FoIs could have been underestimated through the exclusion of DENV infections that were not serotyped. For the second question, it is crucial to consider left-censoring of individual infection history

before study enrollment. An analysis based on the PDCS data but ignoring left-censoring found that the average time from a primary inapparent infection to a second symptomatic infection was longer than that from a primary inapparent infection to a second inapparent infection, suggesting short-term cross-protection[10]. However, the mean times between infections could be underestimated when the analysis is limited to the study observation period.

Coupling the Nicaragua cohort data during 2004–2009 with the national epidemiological surveillance data during 1995–2009 (epidemic year, e.g., 2009, and dengue season, e.g., 2009–2010, are used exchangeably unless otherwise stated.), we develop a statistical framework to answer the above questions while accounting for the uncertainty in infection or serotype status during and before the study. In addition, we also assess the role of pre-dengue-season levels of binding antibodies and neutralizing antibodies in the risk of infection and the risk of subsequent disease given infection.

## Results

**Study participants.** In total, 5086 individuals who contributed definitive information (infection or no infection) for at least one study year were included in the analyses of the Nicaraguan cohort. Summary characteristics of the study participants, together with laboratory test results are provided in Supplementary Tables 1 and 2. The numbers of participants remained similar over the study years (2004–2009). As participants gradually aged within the cohort, the average age of the cohort increased. Based on the partial laboratory confirmation data of the cohort, DENV-1 and DENV-2 co-circulated in the 2004–2005 and 2005–2006 seasons, and then DENV-2 took the lead for the 2006–2007 and 2007–2008 seasons. DENV-3 was the major serotype for the 2008–2009 and 2009–2010 seasons. The outbreaks in the 2005–2006 and 2009–2010 seasons were relatively larger than those in other seasons. The crude probability of symptoms given infection, as represented by the proportion of RT-PCR-confirmed infections among all detected infections, was much higher in the 2009–2010 season, when DENV-3 was dominant, than other seasons. A large proportion of infections was detected solely by the iELISA each year (55–80%) without further serotyping.

**Annual risk of infection.** Based on a Bayesian model fitted to individual data from the cohort and surveillance data, we estimated baseline annual probabilities of infection for each serotype for all seasons from 1995 to 2009, where baseline refers to DENV-naive children ≤ 8 years old in households with home ownership and ownership of 1–2 electric fans (Fig. 1a; Supplementary Table 3). Before the cohort was initiated in 2004, there were four seasons dominated by DENV-3 (1995–1996, 1996–1997, 1997–1998 and 1998–1999), followed by three seasons of primarily DENV-2 (1999–2000, 2000–2001 and 2001–2002) and two seasons of DENV-1 (2002–2003 and 2003–2004). Five seasons reached annual infection probabilities of nearly 10% or more, covering all serotypes except for DENV-4, but all before the study began. The highest infection probabilities were attained by DENV-3 in season 1995–1996, 0.27 (95% credible interval [CI]: 0.23–0.31) and 1998–1999, 0.27 (95% CI: 0.24–0.31). The estimates of baseline annual infection probabilities largely agree with the trajectories of case numbers reported by the surveillance system in Managua (Fig. 1b). The estimation of the infection probabilities during the study period is mainly informed by the study itself. For the years prior to 2004, the estimation is guided not only by surveillance data via the relationship between infection probabilities and surveillance data during the study period

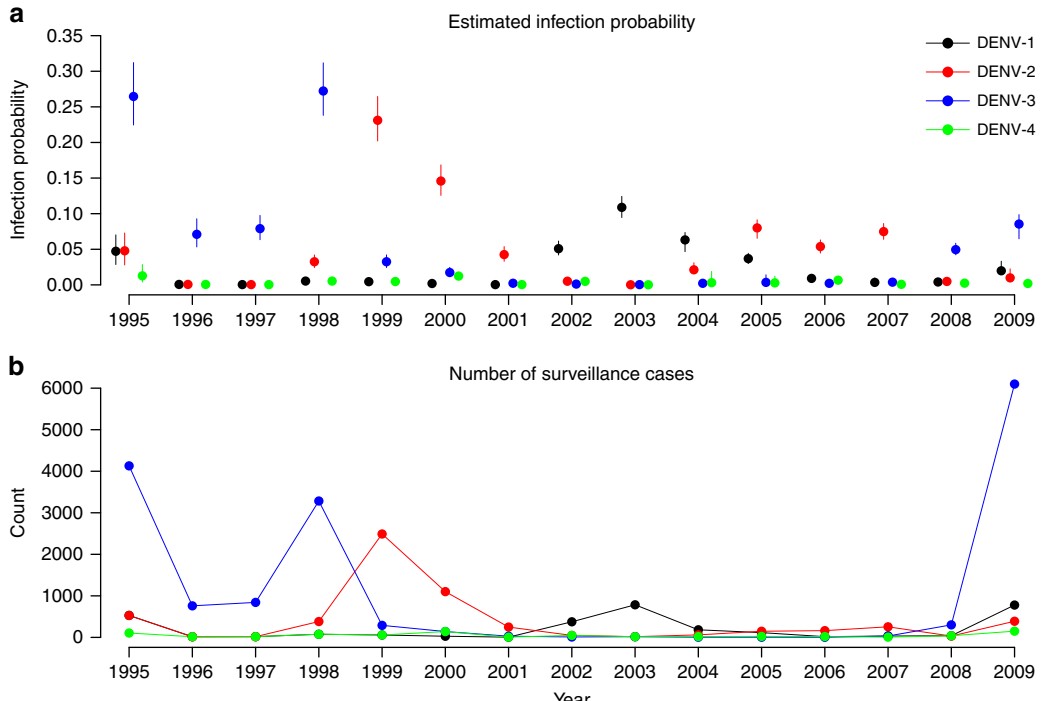

**Fig. 1** Estimated probabilities of infection and reported numbers of cases by serotype and year. Serotypes are colored in black for DENV-1, red for DENV-2, blue for DENV-3, and green for DENV-4. **a** The model-predicted serotype-specific probabilities of infection. Points and vertical bars indicate posterior medians and 95% credible intervals. **b** The number of surveillance-reported cases in Managua, Nicaragua, from 1995 to 2009. The year is defined as July of a year to June of the next year, to be consistent with the dengue season in the study region. Serotype-specific surveillance counts were imputed using information obtained from PAHO and literature (Supplementary Methods). Source data are provided as a Source Data file

but also by observed infection histories during the study period via immunological constraints. This agreement assures us that the information contained in the cohort data during the study period is more or less consistent with that contained in the surveillance counts. In addition, our estimates singled out DENV-2 as imposing much higher risks of infection than other serotypes during 2006–2007 and 2007–2008, which was not obvious in the surveillance data. The relative magnitude of the estimated baseline infection probability for DENV-3 in the 2009–2010 season seems not to match the large outbreak size reported in surveillance data. This is because the baseline probability of infection is for dengue-naive children of 2–8 years old, while the surveillance data reflect the synergy of serotype-specific probabilities of infection, probabilities of disease given infection and all relevant risk factors, e.g., age and infection history. Population growth in Managua during 1995–2009 might have also contributed; that is, the number of DENV-3 cases captured by surveillance in 2009 could be higher than those in the 1990s because of a larger susceptible population, although the risk of infection did not increase.

**Factors affecting risk of infection**. With individual-level infection history sampled via Markov chain Monte Carlo, we were able to assess the association of infection risk with demographics, infection history and living conditions (Fig. 2; Supplementary Table 4, Scenario 2). Compared to children ≤8 years old, older children had a higher risk of infection, with an odds ratio (OR) of 1.55 (95% CI: 1.39–1.74). After controlling for age, having one previous infection significantly reduced the risk of infection with an OR of 0.46 (95% CI: 0.34–0.60), whereas having two or more prior infections increased the risk of infection with an OR of 1.91 (95% CI: 1.34–2.64), both compared to the DENV-naive group.

Among individuals infected once previously, the risk of a secondary infection 3 or more years after the primary infection, compared to 1 year after, was moderately increased with an OR of 1.47 (95% CI: 1.10–2.06), suggesting the potential role of short-term cross-immunity. In contrast, among individuals infected twice or more, the risk of infection exhibited a decreasing tendency over time, with an OR of 0.66 (95% CI: 0.44–1.02) 3 or more years after the most recent infection relative to 1 year after. Indeed, if the risks at 1, 2, and ≥3 years postinfection are compared directly with the DENV-naïve group, one can see a clear trend of decay over time for both the protective effect of the primary infection and the risk-boosting effect of secondary infections: the respective ORs for 1, 2, and ≥3 years are 0.46, 0.52, and 0.68 for the former and 1.91, 1.45, and 1.26 for the latter (Supplementary Notes, Section 2.2). Simple correlation statistics showed that house ownership and possession of electric fans were negatively associated with infection (Supplementary Table 5). Home ownership lowered the risk of infection by about 20%, with an OR of 0.81 (95% CI: 0.71–0.93). Interestingly, having 1–4 electric fans in the house reduced the risk of infection by about 32%, and having 5 or more fans reduced the risk by about 45%, both compared to no fan at all and with statistical significance.

**Factors affecting probability of disease given infection**. The probabilities of disease given infection varied substantially by serotype. DENV-3 was the most pathogenic serotype, with 24% (95% CI: 19–30%) infections being symptomatic (Fig. 3; Supplementary Table 6, Scenario 2). DENV-1 and DENV-2 were similar in their pathogenicity, with the probabilities of disease estimated as 10% (95% CI: 7–13%) and 13% (95% CI: 11–16%), respectively. DENV-4 was the least pathogenic, causing clinical disease in only 2% (95% CI: 0–6%) of infections. Compared to

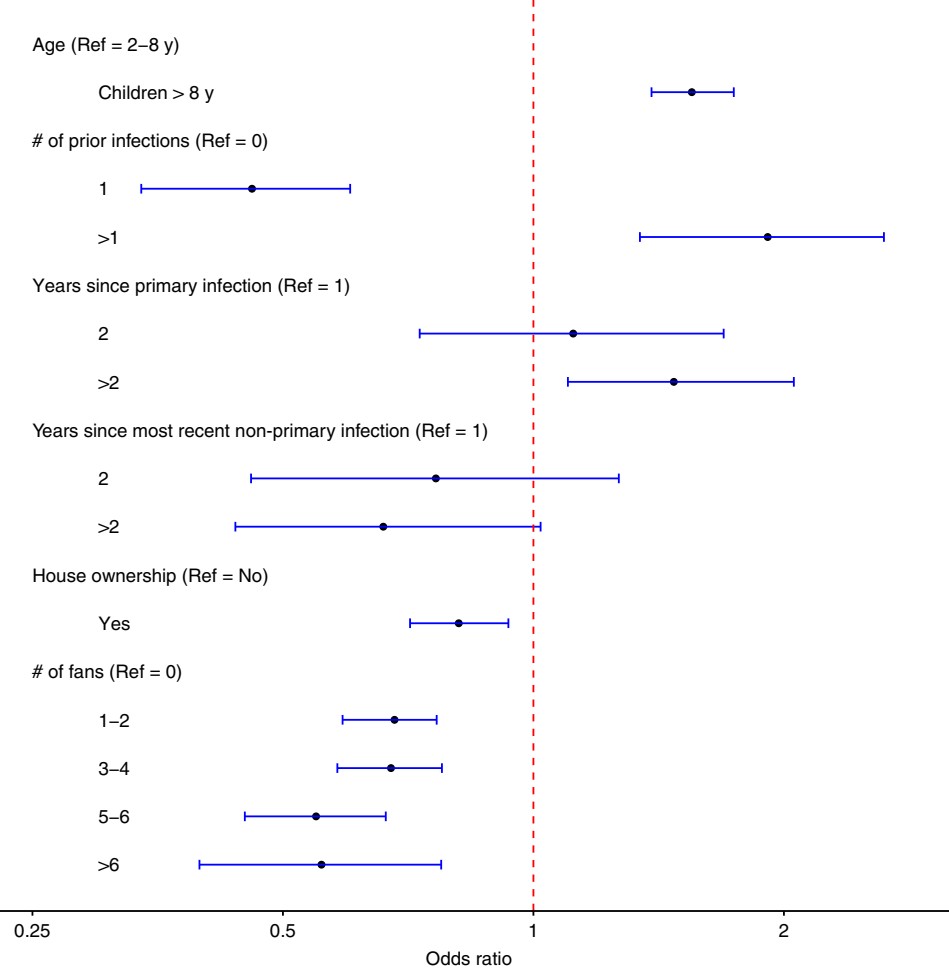

**Fig. 2** Estimated effects of risk factors for DENV infection. Associations with the risk of DENV infection are shown as posterior medians (points) and 95% credible intervals (horizontal bars) of odds ratios for age group, number (#) of prior infections, years since most recent infection, home ownership and number of electric fans. The reference (Ref) group is children aged 2–8 years, without any prior infection, with no household ownership and no fan in their household. Source data are provided as a Source Data file

children ≤8 years old, older children had twice the odds of disease given infection, with an OR of 2.20 (95% CI: 1.46–3.37). We assessed the association between infection history and the probability of disease given infection stratified by age. Among children >8 years old, the OR of disease given infection in year 2 after the most recent infection was 0.19 (95% CI: 0.03–0.77), compared to year 1. We found no association between the number of prior infections and the probability of disease given infection.

**Effects of preseason antibody levels**. We explored the relationship between the preseason total binding antibody levels measured by the nonserotype-specific iELISA and the probabilities of secondary DENV infection, symptomatic secondary infection, and disease given secondary infection (Supplementary Table 7). This analysis was restricted to study person-years with evidence for one or more prior infections. Compared to those with undetectable titers (<10), low-to-medium levels of iELISA titers were associated with increased risk of secondary infection, with ORs estimated as 3.13 (95% CI: 2.16–4.52) for titers 10–20 and 1.83 (95% CI: 1.28–2.62) for titers 21–80. High levels of antibody titers (>1280) were associated with substantially lower risk of secondary infection, with an estimated OR of 0.38 (95% CI: 0.24–0.60). The risk of symptomatic secondary infection was more than twofold for low-to-medium levels of iELISA titers (10–80) as compared to undetectable, but the differences did not

attain statistical significance. For the risk of disease given secondary infection, the highest level (>1280) seemed to be associated with elevated pathogenicity compared to undetectable, with an estimated OR of 3.60 (95% CI: 0.99–13.09). We found no relationship between the risk of disease given secondary infection and Nabs measured by PRNT or NT assays.

**Distribution of time intervals between infections**. The posterior distributions of the time intervals between two consecutive inapparent infections and from an inapparent infection to a symptomatic infection are compared (Fig. 4). When only the infection pairs that occurred within the study period (2004–2009) were used for estimation, we estimated the posterior mean time interval between two inapparent infections to be 2.11 (95% CI: 1.90–2.31) years, slightly shorter than 2.34 (95% CI: 1.97–2.71) years from an inapparent infection to a symptomatic infection. Without such restriction, i.e., using observed and imputed infections during 1995–2009, the posterior mean of time intervals between two inapparent infections, 4.56 (95% CI: 4.26–4.87) years, remained shorter than that from an inapparent infection to a symptomatic infection, 5.02 (95% CI: 4.32–5.71) years, but both are longer than the estimates restricted to the study period only. This underestimation of the time interval between infections when data are restricted to a short period was seen in the simulation study as well.

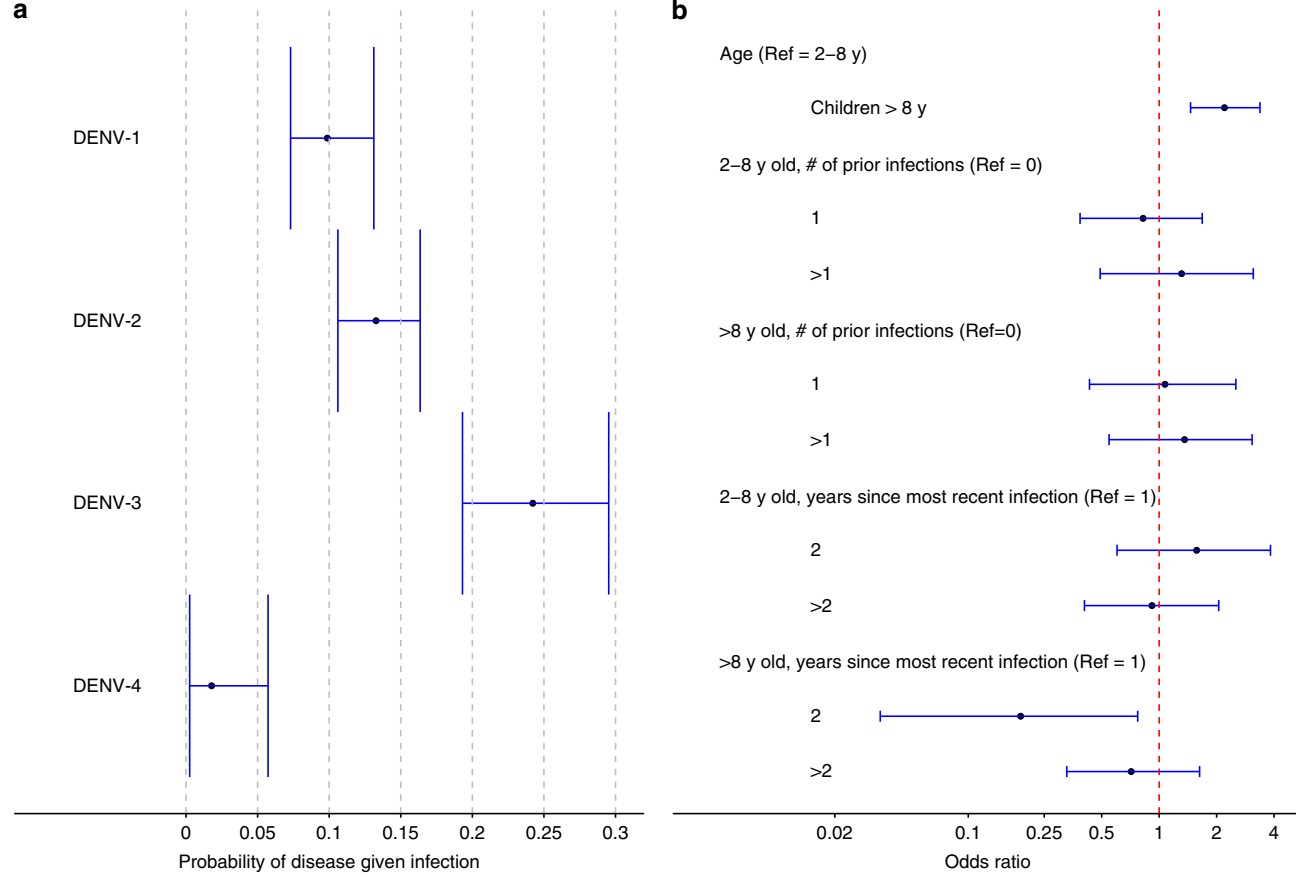

**Fig. 3** Estimated probabilities of disease given infection and effects of risk factors. Points and vertical bars indicate posterior medians and 95% credible intervals. **a** Serotype-specific probability of disease given infection. **b** Association of the risk of disease given infection with serostatus at entry, age group, and years since most recent infection. The reference group is children aged 2–8 years, without prior infection. Source data are provided as a Source Data file

**Sensitivity analyses**. As serotype-specific surveillance data are missing for some years, we conducted sensitivity analyses by changing the proportions of nondominant serotypes in the construction of serotype-specific surveillance data (Supplementary Methods, Section 1.1; Supplementary Figure 1). The proportions of nondominant serotypes are assumed to be 10% in the primary analysis (scenario 2), and 5% (scenario 1) and 15% (scenario 3) in the sensitivity analyses. The estimated annual probabilities of infection, probabilities of disease given infection, and effects of risk factors in the sensitivity analyses are similar to our primary results (Supplementary Figure 2; Supplementary Tables 4 and 6).

**Model adequacy and validation**. By simulating epidemics in the pediatric cohort using the primary model and posterior samples of the parameters, we compared the model-predicted non-serotype-specific annual attack rates to the observed ones (Supplementary Methods, Section 1.7). The similarity between the model-predicted and observed annual attack rates suggests a decent goodness-of-fit of the model to the cohort data (Supplementary Table 9). We also used a conditional expectation approach to test the null hypothesis that the model-predicted infection numbers conditioning on the past and observed infection numbers for each study year were similar (Supplementary Methods, Section 1.7; Supplementary Table 10). The $p$ value of this Chi-squared test was >0.99 for comparison by year and 0.36 if comparison is further stratified by age group, suggesting that the model fits the data satisfactorily.

We further validated our method by repeatedly simulating and analyzing epidemics in a pseudocohort (Supplementary Methods, Section 1.4). Across 100 simulated epidemics, the mean estimates of the parameters were mostly close to, and the interquantile ranges (between 2.5% and 97.5% quantiles of the estimates) all contain, their true values (Supplementary Figure 3), indicating the statistical validity of our method in general. These results together ensure the reliability of our estimates on the annual risks of infection during the study period as well as before the enrollment of the cohort (before 2004).

## Discussion

Understanding the dynamics of DENV infection and risk modifiers requires knowledge about individual-level infection history, which is usually not fully observed even in pediatric cohort studies. To address this challenge, we developed a statistical approach that combines individual-level data from prospective serological cohorts with surveillance data, which facilitates joint inference of missing or left-censored infection history and epidemiological parameters. Using this approach, we estimated serotype-specific annual risk of infection for children in Managua, Nicaragua, from 1995 to 2009. In addition, we assessed risk modifiers for both infection and pathogenicity. In comparison with previous studies that modeled unobserved infections of dengue or influenza, our approach differs in that the complete infection history since birth is sampled for each individual and the sampling is informed by surveillance data[11–13]. We did not model antibody titers dynamics directly, as the serum samples

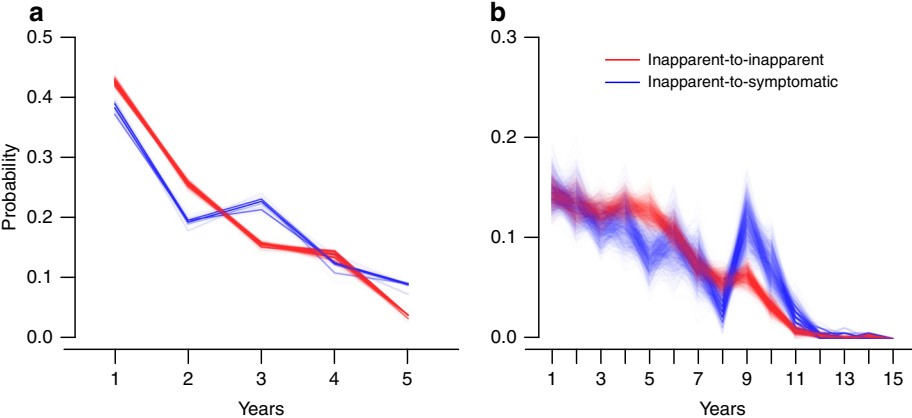

**Fig. 4** Posterior distribution of the time interval between consecutive infections. Results are colored in red for the time between inapparent infections and in blue for the time between an inapparent infection and a symptomatic infection. **a** Estimation is restricted to infections that occurred during the study. **b** Estimation is based on the proposed method combining the study period and the years before study entry. Source data are provided as a Source Data file

were collected at 1-year intervals, in comparison to quarterly collection in Salje et al.[11], and the quantity of serotype-specific antibody titers is limited.

The dramatic fluctuating pattern in the annual probabilities of infection for most DENV serotypes has been previously documented, as many time-varying factors can influence the viral competition for susceptible individuals, including the herd immunity profile, distribution of vectors, climatic conditions, and chance itself[2,9]. Age plays a key role in the risk of DENV infection, especially among children. Older children are more active in exposure-related behaviors, have more body surface area, produce more carbon dioxide, and are hence more likely to be infected as compared to their younger peers; however, their actual risk is complicated by the fact that they have had more prior infections and are more likely protected by cross-immunity[4,14–16]. Without controlling for infection history, a previous study found no difference in the probability of infection between older and younger children in the PDCS cohort[4]. After controlling for infection history, we found that older children (>8 years of age) had a higher risk of infection than younger ones, a difference that might be partially attributable to the lower mobility of the latter[17,18].

We found that one prior infection was protective against but two or more were a risk factor for another DENV infection, and that both effects decayed over time. Cross-immunity may account for the protective effect of a single prior infection[19]. The elevated risk associated with two or more prior infections compared to that among DENV-naive population has been observed before, e.g., in a longitudinal study in Iquitos, Peru, and could be contributed by individuals who were much more exposed to the vectors (mainly *Aedes aegypti*) or more susceptible to DENV and thus repeatedly infected[20,21]. Heterogeneity in susceptibility could be attributed to genetic variation or weaker immune response to prior infections[22–24].

It has long been speculated that a primary DENV infection confers short-term cross-immunity against a secondary symptomatic infection that wanes in less than two years[10,19,25]. Less was known about the duration of cross-immunity against a secondary infection. We found that after the primary infection, the risk of a new infection during the second year was similar to that during the first year. In the third year and onwards, however, the risk was increased by about 47% (Supplementary Table 4). Increasing risk of infection with time since primary infection has not been controlled for in previous analyses of waning cross-protection against symptomatic as compared to inapparent infection and

this may affect estimates of waning cross-protective immunity. After two or more infections, although the risk of a new infection was higher during the first year compared to after a single or no prior infection, there appeared to be a trend of declining risk over the subsequent years. This declining trend could be related to behavioral changes that led to decreased exposure. It may also imply that the cross-immunity after two or more infections becomes more sustained.

Our results showed that home ownership and the ownership of electric fans were associated with lower risk of infection. Previous studies noted that proxies of low social economic status, such as the use of pit latrines and the lack of air-conditioning, were predictive of higher risks of dengue transmission[26,27]. In general, improved living conditions associated with a higher socio-economic status may reduce exposure to mosquitoes. The protective effect of ownership of electric fans is most likely because mosquitoes bite less when fans are blowing, and it could have implications for dengue prevention in resource-limited settings where air-conditioning is not affordable. Even after controlling for these risk-predictive social economic variables, children with two or more prior infections were still identified with a high risk of infection, further indicating the existence of an excessively exposed group.

We found that DENV-3 was the most pathogenic, whereas DENV-4 was the least, among all four serotypes, similar to prospective studies in Thailand[1,11,28]. Previous studies in Nicaragua have found DENV-2 to be associated with severe clinical manifestations[29–31]. A pooled analysis of data from multiple countries identified DENV-1 as the most pathogenic and DENV-2 the least[32]. Together, this suggests the possibility of spatial and temporal heterogeneity in the pathogenicity of DENVs and its determinants. Older children in our study were more than twice as likely to develop disease upon infection, controlling for the number of prior infections, which resembles observations in southern Vietnam[33]. A study among school children in Kamphaeng Phet (KPP), Thailand also suggested seven-year olds experienced a lower symptomatic to inapparent ratio than 8–13 years old children, although that study did not include younger children[1,28].

While secondary infection is a known risk factor for severe manifestations such as DHF, our results suggest that the overall probability of symptomatic disease given infection was invariant to the number of prior infections, regardless of age group (Supplementary Table 6), which is consistent with a previous study on

the PDCS cohort[10]. On the other hand, secondary infections were found more likely to be symptomatic than primary ones in both the multicountry pooled analysis and the study in southern Vietnam[32,33]. However, such difference was not seen in Nicaragua studies included in the pooled analysis, nor was it observed for most serotypes except for DENV-1[32]. The gap between these analyses and ours indicates that the role of prior infections in the risk of disease given infection may vary geographically. We conducted an additional analysis by stratifying serotype-specific probabilities of disease given infection by the number of prior infections (0 vs. ≥1) and found that, for DENV-2, this probability of disease for secondary infections doubled that for primary infections (Supplementary Table 11). The DENV-2 epidemics during the study period were preceded by DENV-1 epidemics during 2002–2004 in Managua (Fig. 1), which could be related to our finding as increased pathogenicity of DENV-2 in terms of severe disease following previous infection with DENV-1 has been noticed before[31,34].

Among children >8 years, there was a strong protection against disease given infection in year 2, in comparison to year 1, after the most recent infection, which extended to year 3 and beyond with diminishing strength (Fig. 3; Supplementary Table 6). In contrast, the probability of disease given infection peaked in year two among younger children, though not significantly higher than the years before and after. The same patterns can also be clearly seen in the crude estimates for the probabilities (Supplementary Table 12). The geometric mean preseason iELISA titers among children >8 years with new infections at 1, 2, and 3 years after the most recent infection were 153, 78, and 59, respectively, vs. 63, 112, and 43 among younger children. There appears to be a positive correlation between the probability of disease given infection and the mean preseason titers. Similar patterns for the effect of years since the most recent infection on the probability of disease given infection are seen if we stratify the effect by the number of prior infections rather than age group (Supplementary Table 13), which is not surprising as older children tended to have more prior infections. These results also suggest pathogenicity may not be irrelevant to the number of prior infections if controlling for the year since the most recent infection, e.g., the ratio between the probability of disease given infection in year 1 after a secondary infection and that after a primary infection is $1.97/0.75 = 2.63$, though not statistically significant.

By deconvoluting the risk of dengue disease into the risk of infection and the risk of disease given infection, we found that the preseason antibody level acted differently on infection and disease given infection. Low-to-medium antibody levels were associated with increased risk of secondary infection, whereas high levels were protective. Although antibody-dependent enhancement (ADE) of severe disease among secondary DENV infection is well-known, this antibody-dependent risk of secondary infection has not been previously reported. Based on the same cohort but a longer observation period, Katzelnick et al.[35] showed a clear pattern of ADE for DENV infections with severe disease outcomes, but not for the risk of symptomatic DENV infection. Consistent with this, a recent analysis of the Thailand-KPP cohort during 1998–2003 by Salje et al.[11] showed an ADE pattern for hospitalization and DHF but not for symptomatic DENV infection or DENV infection. In contrast to these studies, our analysis was restricted to secondary infections. Our study did not show a statistically significant ADE pattern for the risk of symptomatic DENV infection either, but among those with at least one prior infection, iELISA titers ≤80 were associated with nearly twice risk of symptomatic DENV infection as compared to undetectable and higher titers (Supplementary Table 7). Several factors may have contributed to the difference in the association of risk of

DENV infection with pre-infection titer between the PDCS cohort and the Thailand-KPP cohort, e.g., differences in circulating DENV genotypes, exposure frequency, and assays for measuring antibody titers (Haemagglutination inhibition was the main assay in the Thai study). On the other hand, both Salje et al. and our study found that high titers were protective against secondary infections (inapparent and symptomatic).

Our analysis found no clear association between antibody levels and the probability of dengue disease given infection, consistent with the recent analysis of the Thailand-KPP study[11]. Interestingly, both analyses implied some degree of enhancement of disease given infection associated with very high antibody levels (Supplementary Table 7; Extended Data Fig. 6 of Salje et al.). For our study, one explanation is that when the preseason titer was already high, inapparent infections might not be able to induce a fourfold increase in iELISA titer and might have been under-detected, which in turn would artificially raise the proportion of disease among all infections. As a sensitivity analysis, we relaxed the definition of infection such that a twofold increase in iELISA also implies infection when the preseason titer was ≥1280. The pathogenicity-enhancing effect of high titers indeed disappeared (Supplementary Table 8). As expected, the protective effect of high titers against infection also vanished. An alternative plausible explanation is that, as a high preseason antibody level is protective against infection, a high viral load may be necessary for a successful infection and thus result in higher pathogenicity. This effect, if true, could partially account for the observation mentioned above that the probability of disease given infection peaked in year 1 since the most recent infection among children >8 years old but in year 2 among younger children (Supplementary Table 6) and the peak pathogenicity seemed to be associated with a higher mean preseason iELISA titer.

A few limitations should be considered for proper interpretation of our results. First, the serotype-specific antibody detection methods, PRNT and NT, were only available for a small subset of serum samples in this study[36,37]. As a result, most inapparent infections were captured by iELISA, which is nonserotype-specific. Second, antibody levels, especially serotype-specific ones, were not directly modeled due to their limited availability despite their importance in the risk of infection and disease severity[11,23,35]. Therefore, we explored the impact of nonserotype-specific antibody levels by correlating them with model outputs, and the uncertainty of model outputs was accounted for (Supplementary Methods, Section 1.6). Moreover, with most infection times known only up to the year for an observation period of 6 years, the model does not capture variations of the risks at finer time scales or in the long run. Should more years of data become available or serological specimens be sampled more frequently in the future, a parametric curve can be fitted for the effect of the time since most recent infection to better characterize the waning of cross-immunity. Finally, while epidemiological surveillance data greatly improve inference on infection history, the analytic results could be sensitive to the accuracy of such data especially for the period not covered by the cohort data. For example, the estimated infection probabilities for DENV-1 in 1995 and for DENV-4 in 2000 vary substantially with different assumptions on how surveillance-reported cases are allocated to serotypes (Supplementary Figure 2).

In conclusion, our analytic framework offers insights on how the risk of DENV infection and the risk of disease given infection are related to the infection history at the individual level, and is readily generalizable to other complex infectious diseases. While prospective cohorts provide the most valuable information about dengue transmission, we emphasize the necessity to improve the coverage and quality of general surveillance systems in dengue-

endemic countries sponsored by governments or nongovernmental organizations. If future dengue vaccine trials are supplemented with high-quality surveillance data, our method could be used to reliably assess the dependency of vaccine safety and efficacy on individual-level infection history.

## Methods

**Study design**. The PDCS study is an ongoing prospective pediatric cohort study conducted in Managua, Nicaragua[4,7,9]. Recruitments were initiated in August 2004 by home visits as well as at the study health center, Health Center Sócrates Flores Vivas (HCSFV). Children from 2 to 9 years old were invited to join the study. During the first 3 years, participating children remained eligible until 12 years old. This age limit for eligibility was extended to 14 years starting year 4. Participants are encouraged to seek free medical services for all illnesses, in particular for febrile illnesses, at HCSFV 24 h a day all year round. Acute and convalescent samples are collected from individuals who meet the 1997 WHO criteria for suspected dengue or have undifferentiated fever. An annual serum sample is also collected from each study participant in each July before the dengue season started. We included in our analysis the data from the first six years of study (August 2004 to July 2010).

**Surveillance data**. Epidemiological surveillance data of reported dengue cases were obtained from the Pan American Health Organization (PAHO, http://www.paho.org). Proportions of the national surveillance cases accounted for by Managua were obtained from Nicaraguan Ministry of Health (Supplementary Methods, Section 1.1).

**Definition of inapparent and symptomatic infections**. RT-PCR, MAC-ELISA, and iELISA were used to confirm and serotype clinical cases for acute DENV infection[7]. All annual samples were evaluated using the iELISA. A subset of individuals with iELISA-positive samples was selected, whose samples further underwent PRNT or a flow cytometry-based NT to measure serotype-specific Nabs. To determine infection and the infecting serotype, we assume the following priority ranking of the four diagnostic methods: RT-PCR, NT, PRNT, and iELISA. For samples tested with only iELISA, the serotype is unknown. Symptomatic infection was defined as dengue-like symptoms with either detection of DENV RNA by RT-PCR, detection of seroconversion by MAC-ELISA or iELISA, or ≥fourfold rise in iELISA titers between acute and convalescent sera. Paired annual sera with seroconversion or a ≥fourfold increase in antibody titers measured by NT or iELISA, or an increase in percent inhibition measured by single-dilution PRNT from <65 to ≥70, with difference greater than or equal to 20, were considered evidence of inapparent infection for individuals without DENV-confirmed febrile episodes. For the 351 symptomatic infections detected in our study period, 138 (39.3%) of them were hospitalized[4].

**Ethics**. The study was approved by the Institutional Review Boards (IRBs) at the University of California, Berkeley, the Nicaraguan Ministry of Health, and the International Vaccine Initiative. Written consent was obtained from a parent or guardian, or if the guardian was illiterate, the consent form was read aloud in the presence of a witness and the guardian's thumbprint was obtained in lieu of a signature, as approved by the IRBs. Verbal assent was obtained from all children aged six years and older.

**Statistical model**. To estimate serotype-specific annual risk of infection and the effects of infection history on the risk of infection and the risk of disease given infection, we developed a Bayesian modeling framework that integrates the prospective cohort and the surveillance data in the study area (Supplementary Methods, Section 1.2). The proposed model has three components that model the infection outcomes, the disease outcomes given infection, and the probabilistic linkage between the total infection numbers in the study cohort and the surveillance data. For the infection component, annual infection probabilities for each serotype were adjusted for demographic variables and infection history via logistic regressions, where the infection outcome was determined assuming competing risks among the four serotypes (Supplementary Methods, Section 1.2.2). Infection history was proxied by number of prior infections and number of years since the most recent infection. To adjust for heterogeneity in individual level exposure, social economic variables including parent education level, floor type of the house, and ownership of house, car, television sets, electric fans, refrigerators, and animals were screened by Spearman correlation with infection incidence during the study period (Supplementary Methods, Section 1.5). Variables exhibiting strong correlation were included in the model. Similarly, the probabilities of developing disease (symptoms) given infection were also adjusted for demographics and infection history via logistic regressions (Supplementary Methods, Section 1.2.3). To inform the imputation of the infection history before 2004, we assumed there was a common fraction of total DENV infections captured by the syndromic surveillance system across years. The linkage component scales up the yearly number of

infections of each serotype in the cohort to the expected number of reported cases in Managua and assigns a Poisson distribution to the actual number of reported cases (Supplementary Methods, Section 1.2.4). A model schematic is shown in Supplementary Figure 4.

**Statistical inference**. Joint statistical inference on parameters of interest, missing infection histories, and missing socioeconomic covariates was implemented with the data-augmented Markov chain Monte Carlo (MCMC) techniques[38]. In each MCMC step, we first updated the model parameters using the random walk Metropolis–Hasting algorithm, and then we updated the infection history for each individual by exhausting all possible sequences of infections (pathways) that were compatible with the observed data. Compatibility with observed data was verified by the following immunological assumptions: (1) each individual could be infected by at most one serotype in each year; and (2) each individual could be infected by a given serotype at most once. The relative likelihoods of all data-compatible pathways were then computed and served as the weights for sampling the missing infection history. Lastly, we updated the missing values in categorical covariates by considering all possible combinations in these values, similar to the imputation of infection history (Supplementary Methods, Section 1.3). This approach is the Bayesian analogue of the pathway expectation-maximization algorithm we developed for vaccine studies[16,39].

**Distribution of time interval between infections**. From the MCMC samples of unobserved infection and disease outcomes, we derived and compared the posterior distribution of the time from an inapparent infection to a symptomatic infection to that between two inapparent infections.

**Effects of preseason antibody on risk of infection or disease**. We explored the effects of preseason antibody levels on the risks of infection, dengue disease (symptomatic infection), and disease given infection using regression models. All regressions were adjusted for age group (2–8 and >8), number of prior infection (1 and >1) and study year. We examined three types of preseason antibody levels: homologous and heterologous Nabs measured by PRNT or NT and the total DENV-specific binding antibodies measured by iELISA. For Nabs, only their effects on the risk of disease given infection were examined. As pathogenic outcomes of secondary infections tend to be more severe, this exploratory analysis was restricted to the study follow-up period after primary infections. The technical details including how to account for uncertainty from both the regression and the MCMC sampling are provided in Supplementary Methods, Section 1.6.

**Model adequacy and validation**. We evaluated the goodness-of-fit of the model to the data during the study period by two approaches (Supplementary Methods, Section 1.7). We first prospectively simulated dengue epidemics in the PDCS using a subset of the posterior samples of the parameters and compared the simulated nonserotype-specific annual attack rates to the observed ones. We then formally tested the difference between observed nonserotype-specific annual infection numbers and model-predicted frequencies conditioning on the past. To ensure the validity and generalizability of our model, we performed a formal simulation study. We simulated 100 dengue epidemics in a pseudocohort of 2000 individuals. For simplicity, all individuals were set to enter the study at the same time with ages at enrollment randomly assigned to 2–5 years, and to be followed for six years. The infection history before study entry for the whole population, as well as the serotype information for 60% of infections during the study period, were set to be missing, similar to the PDCS data. The cohort was assumed to be contained in a larger population for which surveillance counts by serotype were available to inform the overall risk of infection before study entry. We applied our algorithm to estimate the model parameters for each of the simulated epidemics.

**Reporting summary**. Further information on experimental design is available in the Nature Research Reporting Summary linked to this article.

**Code availability**. Computer code (in R and C languages) for conducting simulations and data analysis can be downloaded from https://github.com/timktsang/Nicaragua_dengue.

## Data availability

Data-sharing requests may be directed to the UC Berkeley Center for the Proteciton of Human Subjects at ophs@berkeley.edu, subject to IRB approval at UC Berkeley. Summary data presented in the figures are included in the Source Data file as part of the Supplementary Information. These summary data, together with the study protocol, can also be downloaded from https://github.com/timktsang/Nicaragua_dengue.

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

## Acknowledgements

This research was supported by NIH grant U54-GM111274 (T.K.T., Y.Y., I.M.L., M.E.H., J.S., D.P.R. and S.L.G.), NIH grant R37-AI032042 (I.M.L., M.E.H., J.S. and Y.Y.), the Pediatric Dengue Vaccine Initiative grant VE-1 funded by Bill and Melinda Gates Foundation (E.H. and A.G.), the Dengue Vaccine Initiative grant DV-11-07 (E.H., L.G. and A.G.), and NIH grant P01 AI106695 (E.H., A.G. and L.C.K.). We thank the study team of the Nicaraguan Pediatric Dengue Cohort Study in the Centro de Salud Sócrates Flores Vivas, the Hospital Infantil Manuel de Jesús Rivera, and the Laboratorio Nacional de Virología at the Centro Nacional de Diagnóstico y Referencia of the Nicaraguan Ministry of Health, as well as the Sustainable Sciences Institute. We are also grateful to the study participants and their families.

## Author contributions

Y.Y., T.K.T., I.M.L. and M.E.H. designed the research. E.H., L.G., A.G., L.C.K., G.K. and A.B. conducted the cohort study. Y.Y., T.K.T., L.G., S.L.G., D.P.R. and J.S. collected the surveillance data. T.K.T., Y.Y. and S.L.G. analyzed the data. Y.Y. and T.K.T. wrote the draft manuscript. Y.Y., T.K.T., I.M.L., M.E.H., L.G., A.G., L.C.K. and E.H. finalized the paper.

## Additional information

**Competing interests:** The authors declare no competing interests.

