## [Peer Review File · Nature Communications]

Reviewers' comments:

Reviewer #1 (Remarks to the Author):

The authors make use of a very nice cohort dataset from Nicaragua, applying statistical models to estimate the relationship between unobserved infection history and subsequent infection and disease outcomes. Getting at such unobserved processes is notoriously challenging, and the authors have presented a sensible approach for getting at this problem. However, in my view there are a few areas where the analysis could be strengthened.

I had the following main comments:

- In Figure 1, the authors compare model estimates for infection rates with observed surveillance data as a validation. But if I understand 2.4 of the supplement, the surveillance data is used to infer the left-censored infection history? So it shouldn't be surprising that the two line up pre-2004?
- On page 15, the authors discuss the role of waning cross-protection. They estimate an increase in risk from the 3rd year onwards, but I wonder whether it would be possible to establish a clearer relationship with the model they are using. If responses wane over time, protection should presumably decline monotonically - would it be feasible to explicitly incorporate a function of this type into the model and estimate the likely waning rate?
- The focus on the role of fans and home ownership seems a little odd given that there were multiple comparisons in Table S6 and although the univariable p-values were significant for these two, they were not overwhelmingly so (the evidence in support of a role for school type or mother education was not much weaker).
- On page 18, the authors note the issue with using a 4-fold cutoff for individuals who have higher baseline titres. It's therefore plausible the use of 4-fold rise as a correlate for infection could have influenced conclusions in other ways too. What would the results look like if the authors used a 2-fold cutoff everywhere?
- There have been several recent studies using Bayesian approaches to infer unobserved infections for dengue (DOI: 10.1038/s41586-018-0157-4) and influenza (DOI: 10.1101/330720, 10.1371/journal.pbio.2004974). Unlike the current study, these incorporated titer values as well, so would be worth discussing how the methods described fit in with other work on similar questions. The authors suggest antibody titers weren't modelled because of limited availability, but they feature prominently in the results and discussion.
- It was useful to see a simulation study to check the model's ability to infer infections, but there weren't many measures of model performance when applied to the data, e.g. residuals. There is also likely to be a lot of collinearity between the variables in 2.2 and 2.3 - would it be possible to tease out which of these are actually most predictive of infection and disease risk?
- It would be helpful to include a model schematic in the supplement - there are a lot of different inputs, so it would be good to have a summary of how they fit together.
- I couldn't find a data availability statement. For the analysis to be reproducible, the authors should make the input data for the model available alongside the paper (or at least, what can be published without affecting participant identifiability), and ideally the model code.
- In Figure 4B, what is causing the peak in the distribution at 9 years? Is it an artefact of the timing of the outbreaks?

Minor point:

- On SI page 9, should $\rho * c$ be ρ/c (as this is what appears in the poisson function)?

Reviewer #2 (Remarks to the Author):

The major claims of this paper relate to the risk of infection and the proportion of infections that are estimated to be symptomatic or asymptomatic at different times after different numbers of infections and with different antibody titres.

The estimates of risk of disease given infection are not novel, with multiple previous papers looking at these in this cohort. In addition, these results are also not clearly put in context of other estimates of this risk, for example: Clapham et al. 2017 PLoS NTDs (using aggregated data from multiple cohorts) estimating a change in risk of disease at different times following infection. The risk of disease at different antibody levels is also shown in Katzelnick et al. 2017 Science and though this paper is discussed, it is still not clear how these two analyses differ.

The estimates of the risk of infection given different infection histories and Ab titres is interesting. However for infection histories, I feel the results could be better presented to help the reader understand how this risk changes over time after the different infections.

In general I found the results of this paper presented as currently not very easy to interpret.

Specific comments:

I'm not sure how meaningful it is to present the "crude odds ratio" here when the paper shows much variation in this number for different groups.

For the annual risk of risk of infection- please be clear what data this was calculated using.

Please define a 3-4 year cycle as these cycles are not clear to me from the data shown. DENV2 perhaps has two peaks 3-4 years apart, whereas DENV4 is low all years, there is one peak of DENV1 in 2003, DENV3 peaks in 1998 and then has low transmission until 2009.

Why was the age group cut –off chosen as 8 years old?

Bottom of Page 7: Please be more specific about how population growth might have contributed.

Figure 2: For the "Years since most recent primary infection" (and non-primary) is the "2" meant to be "between 1 and 2" and should the reference be <1 ?

The presentation of the OR for risk of infection given one past infection, doesn't make sense given the fact that it is later shown that this risk varies depending on time since infection. In my opinion makes this hard for the reader to understand what the impact of the time after infection is and leads to conclusions about what happens generally after one infection, instead of at the different times. If the risk is decreased at any point after one infection, but then increased in those who have an infection more than a year ago compared to one year, what does it mean for the overall risk of infection 1 year after infection compared to no infection?

The serotype specific estimates are interesting, but it would be informative to see this broken down by first and second infection too, the estimates here are an amalgamation of infections occurring in multiple types of individuals which may be obscuring different relationships in different groups.

Figure 3: For time since infection, is this also between 1 and 2 years, and less than 1 as in Figure 2? Also the time since infection should be broken down into since primary and since non-primary infection as in Figure 2. This is important because of the different waning of immunity after the different infections.

Sentence that begins: "Distribution..." Please clarify what "restricting the analysis to the study period" means.

Discussion: The first paragraph makes it sound like you have not used the individual level data or present a method that does not need such data. Please rephrase.

Not sure what "both in reference to one year after Infection" means here? The fact that risk of infection decreases after 1 infection is perhaps counter-intuitive, again there is a conflation of multiple time periods here- and the authors have the data to tease this apart- as they should comment on here.

For the sentence beginning: "While secondary infection...." As on my point on Figure 3, I think that this analysis has not taken into account the time since infection which could hide the relationship between number of previous infections and outcome. Once this analysis has been done, please comment on these results here.

Please be more specific about how what new we learn from the antibody titres and infection results here for disease risk compared to those presented in Katzlenick 2018 paper using the same data.

Responses to comments from reviewers

Reviewer #1 (Remarks to the Author):

The authors make use of a very nice cohort dataset from Nicaragua, applying statistical models to estimate the relationship between unobserved infection history and subsequent infection and disease outcomes. Getting at such unobserved processes is notoriously challenging, and the authors have presented a sensible approach for getting at this problem. However, in my view there are a few areas where the analysis could be strengthened.

Response: We thank the reviewer for the encouraging comments.

I had the following main comments:

(1) In Figure 1, the authors compare model estimates for infection rates with observed surveillance data as a validation. But if I understand 2.4 of the supplement, the surveillance data is used to infer the left-censored infection history? So it shouldn't be surprising that the two line up pre-2004?

Response: We agree that the alignment between the two is not totally surprising, but we think the comparison is still meaningful. The estimation of the pre-2004 infection probabilities is partially guided by the pre-2004 surveillance data via the relationship between infection probabilities and surveillance data during the study period (2004-2009), but it is also partially guided by the infection history of the cohort observed during the study period via immunological constraints. In addition, the estimation of covariate effects further complicate the inference about the infection probabilities during the pre-study years. If the model cannot explain either the cohort data or the surveillance data, it is likely to predict unexpected deviation of the trend of infection risks from that of the surveillance counts. This comparison assures us that the information contained in the cohort data during the study period is more or less consistent with that contained in the surveillance counts.

(2) On page 15, the authors discuss the role of waning cross-protection. They

estimate an increase in risk from the 3rd year onwards, but I wonder whether it would be possible to establish a clearer relationship with the model they are using. If responses wane over time, protection should presumably decline monotonically - would it be feasible to explicitly incorporate a function of this type into the model and estimate the likely waning rate?

Response: Thanks for your suggestions. We agree that it would be more informative if a parametric curve can be fitted for the effect of the time since most recent infection. However, with a time unit of year and an observation period of 6 years for the cohort, we lack sufficient data to support a flexible function for the waning of cross-protection, in particular for the trend after secondary infection. Even for primary functions, data are likely too scarce to inform the tail curvature of the function for the third year and onwards. We think the current discrete format with 3 categories, which is more or less equivalent to a two-parameter function, is the most robust way to characterize the general variation pattern of cross-protection. If more years of data become available in the future, we will certainly consider a flexible function to model cross-protection.

(3) The focus on the role of fans and home ownership seems a little odd given that there were multiple comparisons in Table S6 and although the univariable p-values were significant for these two, they were not overwhelmingly so (the evidence in support of a role for school type or mother education was not much weaker).

Response: We included the number of fans and home ownership because they are the only two that reached the predefined cut-off p-value of 0.05. Indeed, the univariate p-values were not adjusted for multiple comparisons. The purpose of this screening analysis is to rank the potential predictors. Mother education was not considered also because only the “university” category differs from other categories in incidence of DENV infection, but this category has a small number of children. We fitted a logistic regression of observed DENV-infection status during the study period on school type, home ownership and number of fans, where private and semi-private were combined for school type, and number of fans was treated as continuous. School type was not significant (p-value=0.5) in the presence of home ownership (p-value=0.014) and number of fans (p-value=0.032). As a result, we think the number of fans and household ownership best represent the social economic variables

that may be predictive of DENV infection. To clarify the rationale of this choice, we added these reasons to the end of Section 4 in the Supporting Information.

(4) On page 18, the authors note the issue with using a 4-fold cutoff for individuals who have higher baseline titres. It's therefore plausible the use of 4-fold rise as a correlate for infection could have influenced conclusions in other ways too. What would the results look like if the authors used a 2-fold cutoff everywhere?

Response: The phenomenon that a high baseline antibody titer may not be boosted much after infection has been seen in many infectious diseases such as influenza and is termed the antibody ceiling effect ^{1,2}. Another influenza study showed that a ceiling effect is rare, if any, for low to moderate baseline titers ³. In our data, a 2-fold iELISA cutoff leads to twice as many infections as a 4-fold cutoff and will likely induce a low specificity of the infection status, in particular for low baseline titers. In the dengue antibody paper (Salje et al.) you suggested⁴, a 2-fold cutoff in the mean serotypic HI titers seems to have more satisfactory combination of sensitivity and specificity than a 4-fold cutoff. However, in our study, the iELISA titer measures the overall non-serotypic antibody level, which is subject to much higher variation than the mean serotypic HI titer (taking an average can shrink the variability substantially). Consequently, we did not pursue additional analyses using a uniform cutoff of 2-fold.

(5) There have been several recent studies using Bayesian approaches to infer unobserved infections for dengue (DOI: 10.1038/s41586-018-0157-4) and influenza (DOI: 10.1101/330720, 10.1371/journal.pbio.2004974). Unlike the current study, these incorporated titer values as well, so would be worth discussing how the methods described fit in with other work on similar questions. The authors suggest antibody titers weren't modelled because of limited availability, but they feature prominently in the results and discussion.

Response: Thank you for suggesting the references. We agree that it would be more informative to model the dynamics of the antibody levels simultaneously. However, we do believe a fine modelling of antibodies should be supported by more data in our setting. Given that we have annual samples for 6 years but children were born up to 10 years

before the study period, we do not think it is feasible to reliably infer the historical antibody levels. In the Thailand-KPP cohort described in Salje *et al.*'s Nature 2018 paper⁴, the quarterly serological sampling is much more frequent than our study, and the serotypic titers further enriched the information available (note that the random effects modelling the antibody dynamic share mean and variances across serotypes in their paper). In addition, in Salje *et al.*'s paper, the effect of infection history on subsequent infection risks was assumed to be fully mediated by antibody level, and therefore it suffices to estimate baseline antibody level rather than to impute the whole pre-study infection history. Ranjeva *et al.* adopted a somewhat similar approach to model dynamics of influenza strain-specific antibodies, except that they sampled the time of the most recent infection to impute the baseline titers⁵. With limited antibody data, we chose the strategy of imputing the whole infection history that is guided by both cohort and surveillance data. We noticed that Kucharski *et al.* (2018) also reconstructed the complete infection history with influenza A H3N2 strains based on cross-sectional or longitudinal strain-specific HI titers together with prior knowledge about the epidemic years of the strains and the location of the strains in the antigenic space⁶. However, the effect of infection history or pre-season antibody levels on subsequent infection risk was not a focus of their paper.

Although we were not able to model the antibody dynamics, our study does share similar important findings with the Thai study. For example, the probability of disease was largely not correlated, with pre-season antibody levels. Interestingly, the somewhat higher probability of disease given infection at very high titers (>1280) in our study also appeared true in the Thai study (empirical probability, \log_2 titer > 6, extended data figure 6), although the two studies used different assays. Our study also has distinct findings, e.g., an ADE pattern for association of the risk of infection with pre-season titer among those with at least one prior infection. We have added these comparisons to the 9th and 10th paragraphs in DISCUSSION (pages 20-21).

(6) It was useful to see a simulation study to check the model's ability to infer infections, but there weren't many measures of model performance when applied to the data, e.g. residuals. There is also likely to be a lot of colinearity between the

variables in 2.2 and 2.3 - would it be possible to tease out which of these are actually most predictive of infection and disease risk?

Response: In the presence of a large amount of missing data, the marginal likelihood of the observed data is intractable. The traditional measures for goodness-of-fit or model comparisons such as DIC are thus difficult to apply. To assure readers of the adequacy of the model in fitting the data, we tested whether the overall difference between observed and model-predicted annual numbers of dengue infections during the study period is statistically significant or not. The model prediction for each study year is conditional on the past. Specifically, for each study year t , we sample parameters and the infection history of each individual up to year $t-1$ from their posterior distributions, based on which we then calculate the expected number of infections in year t . This is done for all study years and all posterior samples. The average of annual total number of expected infections is treated as the model-prediction and compared with observed annual number of infections. A Chi-squared statistic was formulated⁷. The resulted p-values are 0.997 if prediction is stratified by year only and 0.363 if further stratified by age group, suggesting a decent fit. Please see subsection labelled “Model adequacy and validation”, Appendix Section 6 and Table S10 for more details.

Variables in Appendix Section 2.3 represent infection history and important effect modifiers (age group). They are of the primary inferential interest, and it is important to estimate the effect of each one while controlling for or stratified by others, e.g., the effect of the number of prior infections stratified by age group; therefore, all variables in 2.3 were retained in the model. Variables in Appendix Section 2.2 are time-independent household characteristics related to socioeconomic status (SES). Their correlations with the time-dependent variables in Appendix Section 2.3 are minimal. As we stated in Appendix Section 4, the purpose of considering socioeconomic variables is to minimize the impact of possible selection bias, i.e., children with repeated infections might have been more extensively exposed because of low SES. There exists a certain level colinearity among variables in Appendix Sections 2.2, but Spearman correlation coefficients among variables that were most correlated with infection in our screening are actually only mild, all below 0.1. For the reasons for choosing home ownership and number of fans to represent socioeconomic

status in the final model, please refer to our response to your Main Comment (3). We did not perform formal variable selection for the variables in Appendix Section 2.2 because model comparison (variable selection) in the presence of high-dimensional missing/latent data is both technically less-developed and computationally challenging⁸.

(7) It would be helpful to include a model schematic in the supplement - there are lot of different inputs, so it would be good to have a summary of how they fit together.

Response: We agree a model schematic would be helpful and have attached one as Figure S4 in the SI Appendix. Description about this schematic can be found in Section 2.5 of SI and the figure's caption.

(8) I couldn't find a data availability statement. For the analysis to be reproducible, the authors should make the input data for the model available alongside the paper (or at least, what can be published without affecting participant identifiability), and ideally the model code.

Response: We have added a statement about code and data availability in the methods section.

(9) In Figure 4B, what is causing the peak in the distribution at 9 years? Is it an artefact of the timing of the outbreaks?

Response: Yes, we agree it is likely due to the timing of the outbreaks. Based on the surveillance data, there was a large outbreak in 2009 and there were moderate outbreaks in 1998, 1999 and 2000, matching the observation in Figure 4 that the distribution probabilities for 9, 10 and 11 years are somewhat higher than others.

Minor point:

- On SI page 9, should $\rho*c$ be ρ/c (as this is what appears in the Poisson function)?

Response: Sorry for the confusion. Yes, it should be ρ/c , and it has been corrected

accordingly.

References

- 1 Petrie, J. G., Ohmit, S. E., Johnson, E., Cross, R. T. & Monto, A. S. Efficacy Studies of Influenza Vaccines: Effect of End Points Used and Characteristics of Vaccine Failures. *J Infect Dis* **203**, 1309-1315, doi:10.1093/infdis/jir015 (2011).
- 2 Park, J. K. *et al.* Evaluation of Preexisting Anti-Hemagglutinin Stalk Antibody as a Correlate of Protection in a Healthy Volunteer Challenge with Influenza A/H1N1pdm Virus. *Mbio* **9**, doi:ARTN e02284-1710.1128/mBio.02284-17 (2018).
- 3 Freeman, G. *et al.* Quantifying homologous and heterologous antibody titre rises after influenza virus infection. *Epidemiol Infect* **144**, 2306-2316, doi:10.1017/S0950268816000583 (2016).
- 4 Salje, H. *et al.* Reconstruction of antibody dynamics and infection histories to evaluate dengue risk. *Nature* **557**, 719-723, doi: 10.1038/s41586-018-0157-4 (2018).
- 5 Ranjeva, S. *et al.* Age-specific differences in the dynamics of protective immunity to influenza. bioRxiv 330720; doi:10.1101/330720 (2018).
- 6 Kucharski, A.J. *et al.* Timescale of influenza A/H3/N2 antibody dynamics. *PLoS Biology* **16(8)**, e2004974, doi:10.1371/journal.pbio.2004974 (2018)
- 7 Yang Y, Longini, IM and Halloran, ME. Design and Evaluation of Prophylactic Intervention Using Infectious Disease Incidence Data from Close Contact Groups. *Journal Of the Royal Statistical Society, Series C.* **55**, 317-330 (2006).
- 8 Yang, Y, Halloran, ME, Daniels M and Longini, IM. Modeling Competing Infectious Pathogens from a Bayesian Perspective: Application to Influenza Studies with Incomplete Laboratory Results. *Journal of the American Statistical Association* **105**:1310-1322 (2010).

Reviewer #2 (Remarks to the Author):

The major claims of this paper relate to the risk of infection and the proportion of infections that are estimated to be symptomatic or asymptomatic at different times after different numbers of infections and with different antibody titres.

(1). The estimates of risk of disease given infection are not novel, with multiple previous papers looking at these in this cohort. In addition, these results are also not clearly put in context of other estimates of this risk, for example: Clapham et al. 2017 PLoS NTDs (using aggregated data from multiple cohorts) estimating a change in risk of disease at different times following infection. The risk of disease at different antibody levels is also shown in Katzelnick et al. 2017 Science and though this paper is discussed, it is still not clear how these two analyses differ.

Response: We thank the reviewer for bringing these important questions to our attention. The estimation of the risk of disease given infection is not new, but the simultaneous estimation of the effects of number of prior infections, time to most recent infection, age and their interactions is new. We agree that our results need to be put in the context of existing estimates. For this purpose, we have substantial modifications to paragraphs 6, 7, 9 and 10 of DISCUSSION on pages 18-21.

Specifically, in paragraph 6 (page 18), we referred to Clapham et al.'s pooled analysis of multiple countries that found DENV 1 as the most pathogenic and DENV2 the least, suggesting the possibility of spatial and temporal heterogeneity in pathogenicity¹. We also pointed out in paragraph 7 (page 18), although the pooled analysis found secondary infections were more likely to be symptomatic than primary ones, such a difference was not seen in Nicaraguan data included in the analysis, nor was it observed for most serotypes except for DENV 1. We explained that our analysis was adjusted for age but the pooled analysis was not which may partially account for the gap between the two analyses.

We modified the 9th paragraph in DISCUSSION (pages 20) to further clarify the similarities and differences between Katzelnick et al.'s results and ours². Specifically, we stated that "Based on the same cohort but a longer observation period, Katzelnick *et al.* showed a clear

pattern of ADE for DENV infections with severe disease outcomes, but not for the risk of symptomatic DENV infection.” and that “Our study did not show a statistically significant ADE pattern for the risk of symptomatic DENV infection either, but IE titers ≤ 80 was associated with nearly twice risk of symptomatic DENV infection as compared to undetectable and higher titers”. We warned readers about the difference between the two studies, “our analysis was restricted to secondary infections”. In the same paragraph, we further referred to Salje *et al.*'s recent analysis (Nature, 2018)³ of the Thailand cohort of school children in Kamphaeng Phet (KPP) during 1998-2003 showed an ADE pattern for hospitalization and DHF but not for symptomatic DENV infection or DENV infection. Several factors may have contributed to the discrepancy between the Thailand-KPP cohort and the Nicaraguan PDCS , e.g., differences in circulating DENV genotypes, exposure frequency, and assays for measuring antibody titers (Haemagglutination inhibition was the main assay in the Thai study).

In paragraph 10 of DISCUSSION (page 21), we mentioned that both our study and Salje *et al.*'s analysis of the Thailand-KPP cohort found that there was no clear association between antibody levels and the probability of dengue disease given infection³. However, both analyses indicted some degree of enhancement of disease given infection was associated with very high antibody levels (SI Appendix, Table S7 in our manuscript; Extended Data Fig. 6 of Salje *et al.*).

(2). The estimates of the risk of infection given different infection histories and Ab titres is interesting. However for infection histories, I feel the results could be better presented to help the reader understand how this risk changes over time after the different infections. In general I found the results of this paper presented as currently not very easy to interpret.

Response: To facilitate readers' interpretation of our results on how infection history changed the risk of a subsequent infection, we added further explanation to the paragraph subtitled “Factors affecting risk of infection” in RESULTS (page 9). Specifically, we stated that “if the risks at 1, 2 and ≥ 3 years post infection are compared directly with the DENV-

naïve group, one can see a clear trend of decay over time for both the protective effect of the primary infection and the risk-boosting effect of secondary infections: the respective ORs for 1, 2 and ≥ 3 years are 0.46, 0.52 and 0.68 for the former and 1.91, 1.45 and 1.26 for the latter”. We referred readers to the Section 7.2 of the SI Appendix for details on how these numbers were obtained.

Specific comments:

(1). I’m not sure how meaningful it is to present the “crude odds ratio” here when the paper shows much variation in this number for different groups.

Response: We do agree with the reviewer that the crude odds ratios are not very meaningful given that important risk factors such as age group and infection history are not controlled for. We moved these odds ratios from the main text to the SI Appendix (Sec. 7.1) to serve as auxiliary information for the relative virulence of the serotypes.

(2). For the annual risks of infection- please be clear what data this was calculated using.

Response: The estimation of annual risks of infection is based on the individual data from the cohort and the surveillance data. We added “Based on the model fitted to individual data from the cohort and the surveillance data” at the beginning of the paragraph subtitled “Annual risk of infection”. Please also see the newly added model schematic in Figure S4 in the SI Appendix.

(3). Please define a 3-4 year cycle as these cycles are not clear to me from the data shown. DENV2 perhaps has two peaks 3-4 years apart, whereas DENV4 is low all years, there is one peak of DENV1 in 2003, DENV3 peaks in 1998 and then has low transmission until 2009.

Response: We agree the 3-4 year cycle is not really clear. Therefore, we removed that sentence from the manuscript.

(4). Why was the age group cut -off chosen as 8 years old?

Response: We chose the cut-off of 8 years old because dengue vaccine is licensed for

individuals aged with 9 or above

(http://www.who.int/immunization/research/development/dengue_q_and_a/en/) and we hope to provide relevant information for future research on vaccine design or vaccination strategies. We added this explanation to the SI Appendix Section 2.3.

(5). Bottom of Page 7: Please be more specific about how population growth might have contributed.

Response: The case number is given as the attack rate multiplied by the population size. Therefore, for a growing population, an epidemic in 2009 would have caused more cases than those in the nineties even with similar infection probabilities. We added “that is, the number of DENV-3 cases captured by surveillance in 2009 could be higher than those in the nineties because of a larger susceptible population, although the risk of infection did not increase” (bottom of page 8).

(6). Figure 2: For the “Years since most recent primary infection” (and non-primary) is the “2” meant to be “between 1 and 2” and should the reference be <1?

Response: In the study, infections were mostly identified by annual serological samples and thus we cannot pinpoint exact infection time (except for PCR-confirmed ones). Given such uncertainty, we feel it is equally inaccurate to use “<1 year”, “1-2 years” and “>2 years”, as the actual span from infection to any time during the first year post the infection year could be 0-2 years. Even if considering that the dengue peaks occurred during the rainy season (August-January), time can vary from 6 to 18 months. Therefore, we left the description about years since most recent infection as is in the main text, but add the following clarification in the SI Appendix Section 2.2 (page 6):

“Note that ‘one year’ here refers to the epidemic year after the infection year and does not mean exactly one year (12 months) after the day of infection. This is because infections were identified by annual serological samples, and we cannot pinpoint the exact infection time. The actual time between infection and any time the epidemic year after the infection year could vary from 1 to 24 months. Even if we consider that the epidemic season in Nicaragua is generally August-January, the actual time span could still vary from 6-18

months. Similarly, “two years” refers to the second epidemic year after the infection year, and so on.”

In addition, we acknowledged in the limitations of our analysis that “the estimated effects for the time since most the recent infection do not capture variations of the risks at finer time scales” (page 22).

(7). The presentation of the OR for risk of infection given one past infection, doesn't make sense given the fact that it is later shown that this risk varies depending on time since infection. In my opinion makes this hard for the reader to understand what the impact of the time after infection is and leads to conclusions about what happens generally after one infection, instead of at the different times. If the risk is decreased at any point after one infection, but then increased in those who have an infection more than a year ago compared to one year, what does it mean for the overall risk of infection 1 year after infection compared to no infection?

Response: Thank you for pointing this out. We agree the current parameterization, especially with interactions, could make interpretation not so intuitive. On the other hand, conditioning the effect of time after infection on the number of prior infections does offer valuable insights about what to expect for the risk of a subsequent infection after a primary infection vs. after a secondary infection as time passes. To better help readers understand the implications of the model, we translate current estimates for the impact of number of prior infections and time after infections into estimates of relative risks (in terms of odds ratios) after infection in comparison to a common reference, the DENV-naïve group. We added the following to the paragraph labelled “Factors affecting risk of infection” in the RESULTS section (page 9): “Indeed, if the risks at 1, 2 and ≥ 3 years post infection are compared directly with the DENV-naïve group, one can see a clear trend of decay over time for both the protective effect of the primary infection and the risk-boosting effect of secondary infections: the respective ORs for 1, 2 and ≥ 3 years are 0.46, 0.52 and 0.68 for the former and 1.91, 1.45 and 1.26 for the latter”. We also added a subsection 7.2 to the SI Appendix to explain how these numbers were calculated.

(8). The serotype specific estimates are interesting, but it would be informative to

see this broken down by first and second infection too, the estimates here are an amalgamation of infections occurring in multiple types of individuals which may be obscuring different relationships in different groups.

Response: We did an additional analysis stratifying serotypic probabilities of disease given infection by the number of prior infections (0 vs. ≥ 1). We did not see significant difference in the pathogenicity for DENV-1 and DENV-3. However, the pathogenicity of secondary infections with DENV-2 doubled as compared to primary infection with DENV-2. This could be related to that fact that the DENV-2 epidemics during the study period was preceded by DENV-1 epidemics during 2002-2004 in Managua (Fig. 1), as increased pathogenicity of DENV-2 in terms of severe disease following previous infection with DENV-1 has been noticed before. We have presented this result in Table S11 of the SI Appendix and discussed it at the end of the 7th paragraph in DISCUSSION (page 18-19).

(9). Figure 3: For time since infection, is this also between 1 and 2 years, and less than 1 as in Figure 2? Also the time since infection should be broken down into since primary and since non-primary infection as in Figure 2. This is important because of the different waning of immunity after the different infections.

Response: For naming the time points since last infection, please see our response to Specific Comment (6) about Figure 2. We did reanalyze the data by stratifying the effect of time since last infection by the number of prior infections. For simplicity and also due to data limitations, this stratified analysis is not further stratified by age group. What we saw is very similar to the original results stratified by age group, the pathogenicity peaked in year 2 for one prior infection (matching 2-8 years old) and in year 1 for two or more prior infections (matching >8 years old). This similarity is very likely because the effect of age is confounded with that of the number of prior infections, as older children tended to have more prior infections. We presented the results as Table S13 in the SI Appendix and discussed it in the 8th paragraph in DISCUSSION (page 19).

(10). Sentence that begins: “Distribution...” Please clarify what “restricting the analysis to the study period” means.

Response: We rephrased this sentence as “When only the infection pairs that occurred within the study period (2004-2009) were used for estimation” (bottom of page 12).

(11). Discussion: The first paragraph makes it sounds like you have not used the individual level data or present a method that does not need such data. Please rephrase.

Response: Thanks for the suggestion. Our method surely need individual level data. We now stated in the paragraph that “we developed a novel statistical approach that combines individual-level data from prospective serological cohorts with surveillance data”.

(12). Not sure what “both in reference to one year after Infection” means here? The fact that risk of infection decreases after 1 infection is perhaps counter-intuitive, again there is a conflation of multiple time periods here- and the authors have the data to tease this apart- as they should comment on here.

Response: We agree this statement is confusing and thus rephrased the sentence as “We found that one prior infection was protective against but two or more were a risk factor for another DENV infection, and both effects decayed over time” (page 16). Please see our response to Specific Comment (7) where we offered a better interpretation for the dependence of infection risk on the time since most recent infection after primary and secondary infections.

(13). For the sentence beginning: “While secondary infection.....” As on my point on Figure 3, I think that this analysis has not taken into account the time since infection which could hide the relationship between number of previous infections and outcome. Once this analysis has been done, please comment on these results here.

Response: Please see our responses to specific Comments (8) and (9). We have done additional analyses stratifying either serotypic pathogenicity (Table S11) or the effect of time since last infection (Table S13) by the number of prior infections. Stratifying both by the number of prior infections in the same model causes non-convergence of the MCMC chains, likely because of insufficient data for such a complex model. Based on the new analysis in Table S13, we also added the following to the 8th paragraph (page 19): “These

results also suggest pathogenicity may not be irrelevant to the number of prior infections if controlling for the year since the most recent infection, e.g., the ratio between pathogenicity in year 1 after a secondary infection and that after a primary infection is $1.97/0.75=2.63$, though not statistically significant.” (bottom, page 19)

(14). Please be more specific about how what new we learn from the antibody titres and infection results here for disease risk compared to those presented in Katzelnick 2018 paper using the same data.

Response: We have expended our discussion on the comparison between our study and Katzelnick et al.’s 2018 paper. In addition, we also compared our results to Salje *et al.* ’s 2018 paper on the Thailand KPP study during 1998-2003. Please see our response to your Major Comment (1).

References:

- 1 Clapham, H.E. et al. Immune status alters the probability of apparent illness due to dengue virus infection: evidence from a pooled analysis across multiple cohort and cluster studies. *PLoS Negl Trop Dis* **11**, e0005926 (2017).
- 2 Katzelnick, L. C. *et al.* Antibody-dependent enhancement of severe dengue disease in humans. *Science* **358**, 929-932, doi:10.1126/science.aan6836 (2017).
- 3 Salje, H. *et al.* Reconstruction of antibody dynamics and infection histories to evaluate dengue risk. *Nature* **557**, 719-723, doi:10.1038/s41586-018-0157-4 (2018).

REVIEWERS' COMMENTS:

Reviewer #1 (Remarks to the Author):

The reviewers have addressed most of my comments. However, there are still a couple of issues remaining:

- The authors have made some useful points in their response letter, which would be worth including in the manuscript as well. In particular, the description of Figure 1 should make it clear why the comparison is meaningful (1), and it would be helpful to have a sentence or two in the discussion on what sort of data would be required to fit a more detailed waning model (2) - this could help inform future study designs.

- I'll defer to the editor on whether the new data statement satisfies journal requirements. However, in my view it's not acceptable for a modern epidemiological study to just provide an e-mail address and no supporting data at all. There's plenty of evidence that such practices result in limited eventual data sharing (10.1371/journal.pone.0007078, 10.1186/2046-4053-3-107). At the very least, a reader should be able to replot the main figures and tables from a dataset with the summary results, even if the raw data used in the modelling cannot be published for privacy reasons.

Reviewer #2 (Remarks to the Author):

I thank the authors for their thorough consideration of the comments, and revision of the manuscript. I am satisfied that their revisions now mean this paper is suitable for publication.

Responses to Review Comments

REVIEWERS' COMMENTS:

Reviewer #1 (Remarks to the Author):

The reviewers have addressed most of my comments. However, there are still a couple of issues remaining:

- The authors have made some useful points in their response letter, which would be worth including in the manuscript as well. In particular, the description of Figure 1 should make it clear why the comparison is meaningful (1), and it would be helpful to have a sentence or two in the discussion on what sort of data would be required to fit a more detailed waning model (2) - this could help inform future study designs.

Response: To make it clear why the comparison in Figure 1 is meaningful, we have added the following to the paragraph labeled “Annual risk of infection” in the RESULTS section (Page 8, clean version): “The estimation for the infection probabilities during the study period is mainly informed by the study itself. For the years prior to 2004, the estimation is guided not only by surveillance data via the relationship between infection probabilities and surveillance data during the study period but also by observed infection histories during the study period via immunological constraints. This agreement assures us that the information contained in the cohort data during the study period is more or less consistent with that contained in the surveillance counts.”

To discuss what sort of data would be required to fit a more detailed waning model, we have added the following to the limitations in DISCUSSION (page 20): “Moreover, with most infection times only known up to the year for an observation period of 6 years, the model does not capture variations of the risks at finer time scales or in the long run. Should more years of data become available or serological specimens be sampled more frequently in the future, a parametric curve can be fitted for the effect of the time since most recent infection to better characterize the waning of cross-immunity.”

- I'll defer to the editor on whether the new data statement satisfies journal requirements. However, in my view it's not acceptable for a modern epidemiological study to just provide an e-mail address and no supporting data at all. There's plenty of evidence that such practices result in limited eventual data sharing (10.1371/journal.pone.0007078, 10.1186/2046-4053-3-107). At the very least, a reader should be able to replot the main figures and tables from a dataset with the summary results, even if the raw data used in the modelling cannot be published for

privacy reasons.

Response: We are not allowed to make the cohort data publicly available as there are IRB requirements. We have included in the resubmission a Source Data file containing all summary data needed for generating all the relevant figures, as required by the journal. In addition, we have uploaded all the code used for simulation and data analysis, together with the Source Data file, to a public repository https://github.com/timktsang/Nicaragua_dengue.

Reviewer #2 (Remarks to the Author):

I thank the authors for their thorough consideration of the comments, and revision of the manuscript. I am satisfied that their revisions now mean this paper is suitable for publication

Response: We thank the reviewer for his or her encouraging comments.